

# Mott-insulator-aided detection of ultra-narrow Feshbach resonances

**Manfred J. Mark[1,2]⋆, Florian Meinert[3], Katharina Lauber[1] and Hans-Christoph Nägerl[1]**

**1** Institut für Experimentalphysik und Zentrum für Quantenphysik,
Universität Innsbruck, 6020 Innsbruck, Austria
**2** Institut für Quantenoptik und Quanteninformation,
Österreichische Akademie der Wissenschaften, 6020 Innsbruck, Austria
**3** 5. Physikalisches Institut and Center for Integrated Quantum Science and Technology,
Universität Stuttgart, Pfaffenwaldring 57, 70569 Stuttgart, Germany

⋆ manfred.mark@uibk.ac.at

## Abstract

We report on the detection of extremely narrow Feshbach resonances by employing a Mott-insulating state for cesium atoms in a three-dimensional optical lattice. The Mott insulator protects the atomic ensemble from high background three-body losses in a magnetic field region where a broad Efimov resonance otherwise dominates the atom loss in bulk samples. Our technique reveals three ultra-narrow and previously unobserved Feshbach resonances in this region with widths below $\approx 10\,\mu$G, measured via Landau-Zener-type molecule formation and confirmed by theoretical predictions. For comparatively broader resonances we find a lattice-induced substructure in the respective atom-loss feature due to the interplay of tunneling and strong particle interactions. Our results provide a powerful tool to identify and characterize narrow scattering resonances, particularly in systems with complex Feshbach spectra. The observed ultra-narrow Feshbach resonances could be interesting candidates for precision measurements.


# 1 Introduction

Ultracold atomic gases provide an ideal platform for precise studies of atom-atom interactions in the low-temperature regime of quantum scattering. As such, they allow to gather experimental information on the fine details of interaction potentials for benchmarking theoretical models that are in most cases beyond the scope of ab initio calculations. Accurate knowledge of atom-atom interactions, in turn, forms the basis for exploiting quantum gases as tunable quantum simulators [1], examples being the celebrated Fermi- and Bose-Hubbard system [2–5], interacting dipolar gases [6], or charge-neutral hybrids [7]. Precision spectroscopy, such as performed in optical lattice clocks, also depends on an accurate determination of particle interactions [8,9]. An important experimental contribution to pin down interaction potentials with highest accuracy constitutes the characterization of Feshbach scattering resonances, i.e. their positions and widths [10–13]. Moreover, very narrow resonances have even been proposed for ultracold atom-based detection of changes in fundamental constants [14–16].

In systems with a rich Feshbach spectrum, one often encounters broad and narrow overlapping Feshbach resonances. This can lead to situations where narrow resonance features are completely disguised by broader ones and are hard to detect. Here, we report on an optical-lattice-based method that allows us to observe such hidden Feshbach resonances for the example of ultracold cesium atoms in the vicinity of a broad Efimov three-body loss resonance [17]. Our approach leads to strong suppression of the Efimov loss feature, which fully dominates the loss in the absence of the lattice, by protecting the sample in a Mott-insulating state. This provides access to previously unobserved ultra-narrow $g$-wave Feshbach resonances with widths on the order of $\approx 10\,\mu$G in the low magnetic field region $B < 10$ G. We measure the resonance positions via Feshbach spectroscopy in the strongly interacting system and provide a detailed characterization of their widths by analyzing the efficiency of Feshbach-molecule production. Finally, we study in detail the loss features caused by the resonances, which in the lattice result from resonant tunneling processes [18,19].

# 2 Feshbach spectroscopy

Experimental detection of Feshbach resonances in trapped atomic ensembles typically relies on the observation of enhanced atom loss in the vicinity of the resonance pole as a result of the rapid increase of the three-body recombination rate with the diverging scattering length $a_s$. Consequently, the overlap of multiple recombination-induced loss features can hinder the identification of Feshbach resonances, in particular, when they are narrow and arise from a comparatively weak coupling to bound molecular states. In a first set of measurements, we study such a scenario for an ultracold cesium sample and explore Feshbach spectroscopy for a strongly correlated lattice gas as a pathway to observe otherwise hidden Feshbach resonances.

We start the discussion with standard Feshbach spectroscopy on a weakly confined thermal, ultracold ensemble by measuring magnetic-field dependent atom loss. Specifically, our experiment starts with $\approx 3 \times 10^5$ cesium atoms in the lowest hyperfine state $|F = 3; m_F = 3\rangle$ trapped in a crossed optical dipole trap and prepared at a temperature $T = 137(10)$ nK. During sample preparation, we apply a magnetic offset field $B$ of about 21 G, for which $a_s = 210\,a_0$. Additionally, a magnetic field gradient of 31 G/cm along the vertical $z$-axis compensates the gravitational force and thus levitates the trapped ensemble [20]. We ramp $B$ with a speed of $\sim 2.5$ G/ms to a value in the range from 2 to 17 G and measure the remaining atom number $n_A$ after a fixed hold time of $t_H = 50$ ms.

The measurement (Fig.1(a)) reveals multiple loss features. Three resonances at 15 , 14.3 , and 11 G are clearly visible and are attributed to previously detected Feshbach resonances

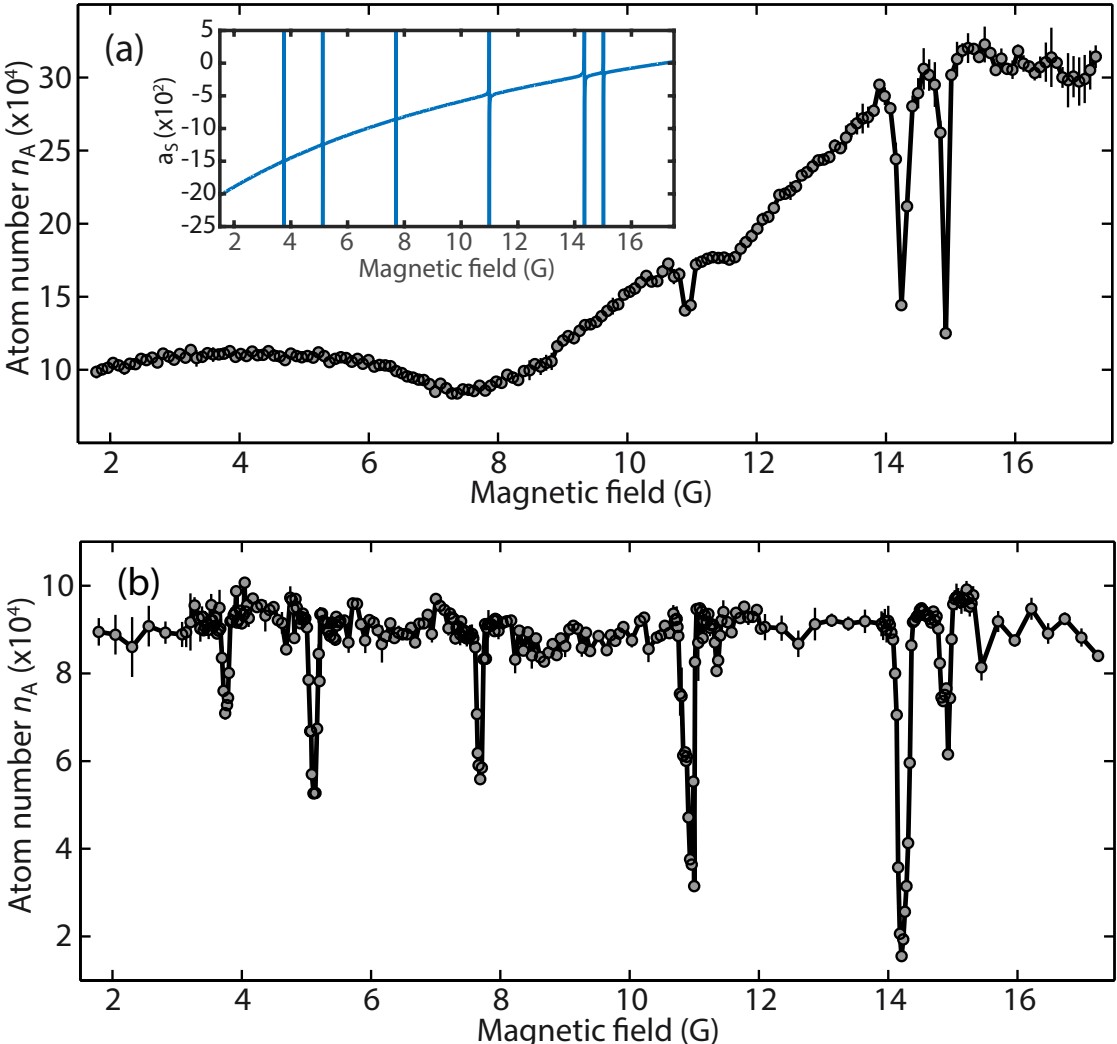

Figure 1: Feshbach atom-loss spectroscopy with and without the optical lattice. Remaining atom number $n_A$ as a function of magnetic field $B$ for a thermal ensemble ($T = 137(10)$ nK) after a hold time $t_H = 50$ ms (a) and for a Mott insulator prepared at $V_{x,y,z} = 20\,E_R$ after $t_H = 200$ ms (b). Three Feshbach resonances above 10 G are evident in both spectra. Ultra-narrow resonances below 10 G are only observed in the optical lattice. They are masked by a broad Efimov resonance around 8 G [17] in the bulk system. Weakly increased loss at 8.8 and 11.5 G in (a) is due to four-body Efimov states [21]. Inset in (a): Calculated scattering length $a_s$ as a function of $B$ showing the position of predicted Feshbach resonances in the magnetic field region of interest. Error bars indicate one standard deviation and solid lines connect the data points to guide the eye.

arising from the coupling to the molecular states $f\,l(m_f) = 6g(5)$, $4g(3)$, and $4g(2)$, respectively [10]. Here, $f$ is the total internal angular momentum with its projection $m_f$, and $l$ denotes the nuclear rotational angular momentum. Note that $M_m = m_f + m_l = 6$ for the projection $M_m$ of the molecule's total angular momentum $\vec{F}_m = \vec{f} + \vec{l}$. For decreasing values of $B$ the measurement is dominated by a broad loss feature centered around 8 G, which is connected to an Efimov trimer resonance [17]. Weak signatures of four-body Efimov resonances at 8.8 and 11.5 G [21] are also visible. The measurement shows that the strong and

broad three-body loss signature arising from coupling to the Efimov trimer completely masks potential Feshbach resonances below 10 G. Indeed, narrow Feshbach resonances have been predicted in this region (see inset to Fig.1(a) for the calculated scattering length based on multichannel calculations [22]) but have so far been lacking observation due to the dominant Efimov resonance.

Next, we demonstrate the first detection of these narrow Feshbach resonances by starting from a Mott-insulating state in an optical lattice at low filling fraction. For this, evaporative cooling is continued until we obtain a Bose-Einstein condensate (BEC) of $1.0 \times 10^5$ atoms [20]. The sample is then adiabatically loaded into a cubic optical lattice with a final lattice depth of $V_{x,y,z} = 20\,E_R$, where $E_R = h^2/(2m\lambda^2)$ denotes the recoil energy with the mass $m$ of the Cs atom and $\lambda = 1064.5$ nm the wavelength of the lattice light. The loading procedure results in a Mott insulator of predominantly singly occupied lattice sites and a residual ($\sim 10$ to $20\,\%$ of the total atom number) double occupancy in the center. Subsequently, we ramp $B$ to the desired value as before and hold the sample for 200 ms. To avoid any loss or heating when ramping over the already known Feshbach resonances, we use ultrafast ramps with a speed of $2 \cdot 10^4$ G/ms across those field regions [23]. As the initial Mott insulator is always prepared at repulsive interaction and subsequently quenched to the vicinity of the Feshbach resonance of interest for the loss spectroscopy, the correlated many-body state features lifetimes above 10 s also for large negative values of $a_s$ [24].

After the hold time we ramp the field quickly (ramp speed $\sim 25$ G/ms) to $\approx 18.5$ G to prevent further atom loss during the subsequent release of the atoms from the lattice and the dipole trap beams (within 2 ms). Finally, we allow for 20 ms time-of-flight expansion before detecting the atom number by standard absorption imaging. The result of this measurement is shown in Fig. 1(b). Evidently, the previously dominant Efimov loss resonance observed without the lattice is now absent, indicating the strong suppression of three-body recombination in the Mott insulator as a result of atom number squeezing. This allows us to identify three additional and so far unobserved resonances in the field region below 10 G. They are located near 7.7, 5.1, and 3.7 G in agreement with theoretical predictions [22] and arise from coupling to the molecular states $f l(m_f) = 6g(4), 6g(3),$ and $6g(2)$, respectively [10].

## 3   Resonance width via molecule formation

Before investigating the loss features in the lattice in more detail, we first measure the widths of the narrow Feshbach resonances in a way that does not rely on the shape of the loss feature and is largely insensitive to residual magnetic field fluctuations. For this, the efficiency of Feshbach molecule association via a magnetic-field sweep across the resonance is monitored as a function of the sweep rate $\dot{B}$ [25]. For an initially doubly occupied lattice site, the molecule formation probability depends on $\dot{B}$ and is described via the well-known Landau-Zener formula [26]

$$p = p_0 + (1 - p_0)e^{-2\pi\delta_{\text{LZ}}}. \tag{1}$$

Here, $p$ denotes the probability for the initial atom pair to remain unbound after the field sweep. The dependence on the sweep rate is encoded in

$$\delta_{\text{LZ}} = \frac{\sqrt{6}\hbar}{\pi m a_{\text{ho}}^3} \left| \frac{a_{\text{bg}}\Delta B}{\dot{B}} \right|, \tag{2}$$

with $\Delta B = B_0 - B^*$ the width of the Feshbach resonance, defined by the magnetic field values of the resonance pole $B_0$ and the zero crossing $B^*$ of the scattering length $a_s$. Further, $a_{\text{bg}}$ is the background scattering length, $a_{\text{ho}}$ the harmonic oscillator length for atoms confined at

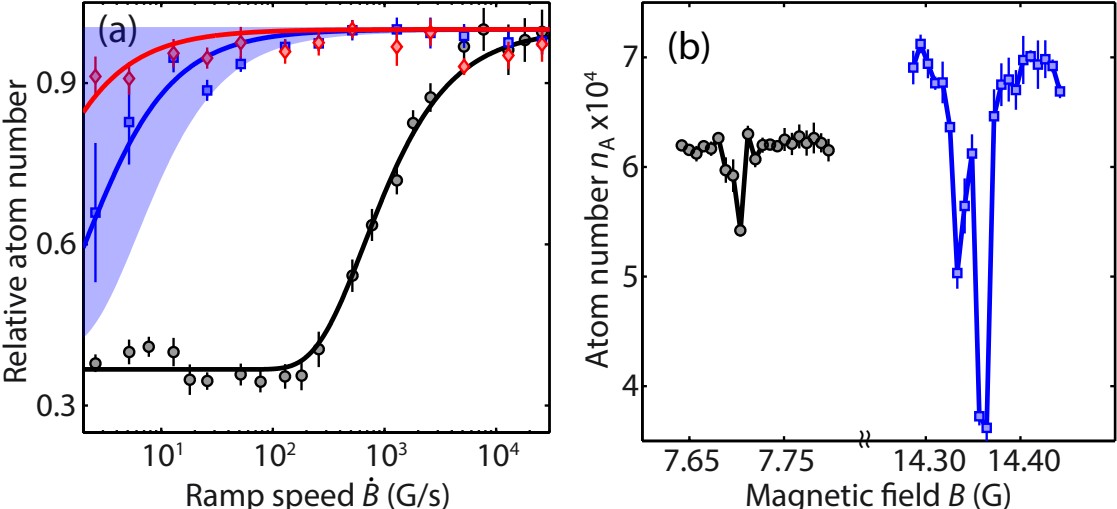

Figure 2: (a) Relative atom number after ramping across the Feshbach resonance at 14.3 G (circles), 7.7 G (squares) and 3.7 G (diamonds) as a function of the ramp speed $\dot{B}$. Here, the lattice depth $V_{x,y,z} = 20\,E_R$ ($V_{x,y,z} = 30\,E_R$) for the resonance at 14.3 G (7.7 G and 3.7 G). Solid lines are fits to the data using Eq. 1 with $\Delta B$ as free parameter, giving $\Delta B = 14.0(0.9)\,$mG (circles), $\Delta B = 8.0(1.6)\,\mu$G (squares), and $\Delta B = 1.2(0.4)\,\mu$G (diamonds). The shaded area indicates the uncertainty region $0\,\mu$G $< \Delta B < 20\,\mu$G for the 7.7 G resonance as explained in the text. (b) Remaining atom number $n_A$ starting from an initially singly occupied Mott insulator state at $V_{x,y,z} = 20\,E_R$ as a function of $B$ around 7.7 G with $t_H = 5000\,$ms (circles) and around 14.3 G with $t_H = 50\,$ms (squares). Solid lines connect the data points and are guides to the eye.

a lattice site, and $p_0$ a constant offset that accounts for residual single occupancy and finite maximum conversion efficiency.

Our measurements of $\Delta B$ for the individual resonances start with the preparation of a Mott insulator with predominant double occupancy. Briefly, this is achieved by applying a stronger external harmonic confinement during lattice loading to maximize the number of doubly occupied lattice sites. The sample is then purified by selective removal of singly occupied sites. This cleaning sequence exploits a Feshbach resonance at 19.9 G to transfer doublons into molecules, before a combined radio-frequency and resonant-light pulse is applied to remove the singly occupied sites. Finally, the molecules are transferred back into unbound atom pairs (for details see Ref. [27]). Note that during the whole sequence the magnetic gradient field is switched off to achieve maximum conversion efficiency.

After sample preparation, the magnetic field is quickly set to a value $\sim 0.5\,$G above the Feshbach resonance of interest. This is the starting point for a linear sweep of $B$, which stops $\sim 0.5\,$G below the resonance. The remaining number of atoms that have not been associated into molecules via the field sweep is then measured as a function of $\dot{B}$. Exemplary datasets for the resonances located at 14.3, 7.7, and 3.7 G are shown in Fig. 2(a). A fit of the model, Eq. 1, to the data allows to directly extract $\Delta B$ for the comparatively broad Feshbach resonances at 15, 14.3, and 11 G. Here the lattice depth was chosen to be $V_{x,y,z} = 20\,E_R$. For the narrow Feshbach resonances below 10 G the situation is more delicate, as we will discuss in the following paragraph.

A smaller resonance width requires lower sweep rates for efficient molecule formation. In our experiment, the minimum useable ramp speed is limited by low-frequency magnetic

field noise on the order of 10 mG (peak-to-peak value) with main spectral components at 50 Hz and 150 Hz. Specifically, this leads to shot-to-shot fluctuations of the sweep rate over the resonance on the order of $\approx 2$ G/s, causing strong fluctuations in the converted number of molecules for sweep rates below a few G/s. To partially circumvent this limitation, we use the fact that based on Eq. 2 the minimum ramp speed for partial conversion can be shifted upwards by decreasing the harmonic oscillator length $a_{\mathrm{ho}}$. For the measurements of the narrow resonances below 10 G, we thus increased the lattice depth to $V_{x,y,z} = 30 E_R$ to shift the signal to faster ramp speeds. As shot-to-shot fluctuations are still relatively strong for low ramp speeds (*cf.* Fig. 2(a)), we account for this by an additional uncertainty region of $0-20 \,\mu$G, exemplary indicated by the shaded area in Fig. 2(a) for the case of the 7.7 G Feshbach resonance. For completeness, we have also measured the width of the resonance at 19.9 G used for the sample purification as described above (not shown in Fig. 1). All results are summarized in Table 1. The experimentally deduced widths are in very good agreement to recent coupled-channels calculations [28]. Even for the most narrow Feshbach resonance with a calculated width of $2 \,\mu$G the experimental value of $1.2(0.4) \,\mu$G obtained from the fit to the data based on Eq. 1 is in good agreement.

## 4 Loss structure in the optical lattice

We now perform a more detailed analysis of the atom-loss structure at the Feshbach resonances when measured in the correlated lattice system. This is not only of relevance for a precise determination of their positions, but also reveals fine details on the loss mechanism in the lattice. First, we note that the loss features in Fig. 1 are inhomogeneously broadened by the applied magnetic-field gradient that counteracts gravity. Once the sample is prepared deep in the Mott-insulating phase, however, the gradient field can be removed during the loss measurements as the lattice is strong enough to hold the atoms against gravity. Second, we observe that the residual double occupancy leads to a broadening and enhancement of the loss features. Possible mechanisms leading to stronger heating and losses in this situation include on-site molecule formation, coupling of atom pairs to higher lattice bands in the vicinity of the Feshbach resonance, and comparably larger off-site three-body losses [29]. Therefore, we modify our loading procedure into the Mott insulator state such that we end up with essentially pure single occupancy as detailed in Ref. [30].

This allows us to perform high-resolution atom-loss spectroscopy in the close vicinity of all the observed resonances that are shown in Fig. 1. In Fig. 2(b), we show two examples for magnetic-field scans around the resonances at 7.7 and 14.3 G at a lattice depth $V_{x,y,z} = 20 E_R$. The hold time for the Feshbach resonance at 7.7 G has to be increased to 5 s to be able to resolve this loss feature. This is a consequence of the combination of the ultra-narrow width and the presence of magnetic-field noise, which leads to a periodic sampling of the magnetic field values around the set value with an amplitude on the order of 10 mG (peak-to-peak value). Therefore, the effective time on resonance is drastically reduced, as visible in the reduced loss. The ultra-narrow resonance at 7.7 G still remains a single loss feature with a significantly reduced width almost down to the resolution limit of our experimental magnetic field control ($\approx 8$ mG step size). In contrast, the narrow Feshbach resonance at 14.3 G reveals a clear substructure by exhibiting two distinct loss features.

The origin of the split resonance structure can be found by varying the lattice depth $V_{x,y,z}$ at which the loss measurements in the Mott insulator state are carried out. Measurements of the split loss feature for $V_{x,y,z} = 20 E_R$ and $V_{x,y,z} = 30 E_R$ are shown in Fig. 3 for the example of the Feshbach resonance at 19.9 G. Evidently, the dip at higher magnetic field is rather insensitive to the variation of the lattice depth, while the dip at lower magnetic field shifts towards smaller

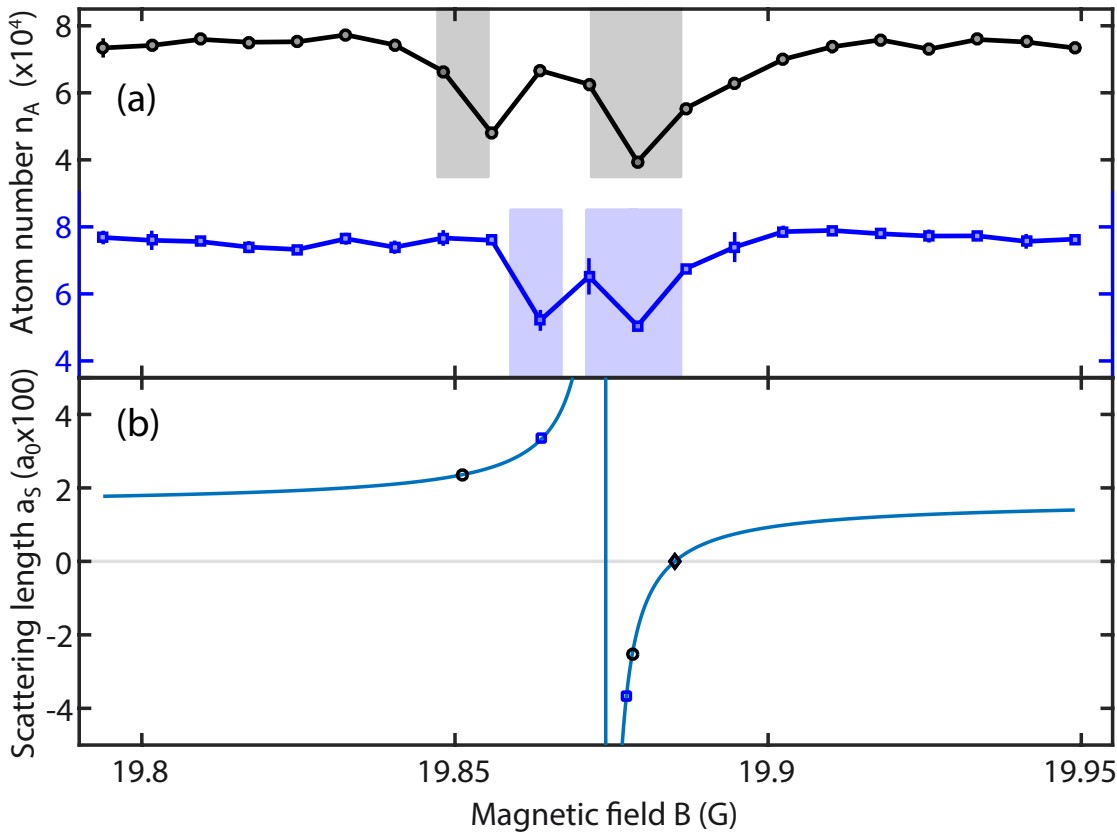

Figure 3: (a) Remaining atom number $n_A$ for a singly occupied Mott insulator as a function of $B$ in the vicinity of the Feshbach resonance at 19.9 G with $t_H = 500$ ms and $V_{x,y,z} = 30\,E_R$ (top circles) and $t_H = 50$ ms and $V_{x,y,z} = 20\,E_R$ (bottom squares). The shaded areas mark the regions for which $|U| = E$, incorporating a width of 8 mG (see (b)). The solid lines connect the data points are guides to the eye. (b) Calculated scattering length around the Feshbach resonance using $B_0 = 19.874$ G and $\Delta B = 11$ mG. The local background scattering length in this range is $\approx 160\,a_0$. Magnetic field values where $|U| = E$ for $V_{x,y,z} = 30\,E_R$ (circles) and $V_{x,y,z} = 20\,E_R$ (squares), and where $|U| = 0$ (diamond) are indicated.

values of $B$ for increasing $V_{x,y,z}$. These observations are a result of the variation of the on-site interaction $U$ near the Feshbach resonance. Starting from a one-atom-per-site Mott insulator, particle loss requires atom tunneling in combination with on-site three-body recombination. To first order, this is possible when $U$ is sufficiently small, as it is the case in the vicinity of the zero crossing of the Feshbach resonance $B^*$. Accordingly, a particle-loss feature arising from such a mechanism is expected to be rather independent of the lattice depth. Alternatively, the removal of the magnetic gradient field subjects the sample to gravity, which introduces an energy offset $E$ between neighboring lattice sites along the vertical $z$-direction. This provides a second loss channel, which occurs at a magnetic field where $|U| \approx E$. Here, resonant tunneling along $z$ becomes energetically allowed [18, 19], and thus opens up the system again to three-body recombination. Evidently, the associated loss features should shift with varying lattice depth as $U$ increases for increasing $V_{x,y,z}$ [31].

To test our reasoning, we plot the calculated scattering length around the Feshbach resonance according to its functional dependence on the magnetic field $a_s(B) = a_{bg} \cdot (1 - \Delta B/(B - B_0))$ in Fig. 3(b) [11]. For the modeling we neglect corrections due to finite range and finite collision energy [32, 33] as well as corrections due to

Table 1: Experimentally determined and theoretically predicted Feshbach resonance positions and widths [28]. Note that the values and error bars for resonances $< 10\,\text{G}$ are the bare fit values. We estimate an uncertainty region of $0 - 20\,\mu\text{G}$ for those resonances.

| Resonance | Experiment | | Theory | |
|---|---|---|---|---|
| | $B_0(G)$ | $\Delta B(mG)$ | $B_0(G)$ | $\Delta B(mG)$ |
| 4g(4) | 19.874(0.01) | 11.1(0.7) | 19.682 | 9.7 |
| 6g(5) | 15.014(0.01) | 3.4(1.0) | 14.761 | 3.6 |
| 4g(3) | 14.345(0.01) | 14.0(0.9) | 14.195 | 13.4 |
| 4g(2) | 10.994(0.01) | 3.2(0.3) | 10.893 | 6.0 |
| 6g(4) | 7.704(0.01) | 0.0080(0.0016) | 7.555 | 0.016 |
| 6g(3) | 5.122(0.01) | 0.0014(0.0008) | 5.038 | 0.006 |
| 6g(2) | 3.753(0.01) | 0.0012(0.0004) | 3.703 | 0.002 |

higher bands [30]. The background scattering length $a_{\text{bg}}$ is known from multi-channel calculations [22] and the resonance width (in this case $\Delta B = 11.1\,\text{mG}$) is fixed from the independent measurement above based on the molecule formation. When using $B_0 = 19.874\,\text{G}$ for the pole of the resonance the observed loss features are in good agreement with the tunneling-induced decay mechanism discussed above. This is indicated by the symbols in Fig. 3(b), which denote the magnetic field values where $U = 0$ (diamond) or $|U| \approx E$ (circles and squares) for the two different lattice depths. Additionally, we mark the magnetic field values where $|U| = E$ by the shaded areas in Fig. 3(a) with a width that indicates our typical magnetic field noise. Note that the two expected loss channels for which $U = -E$ and $U = 0$ are not resolvable given our experimental resolution and hence lead to a single loss feature.

A similar analysis is performed for the narrow Feshbach resonances at $15\,\text{G}$, $14.3\,\text{G}$ and $11\,\text{G}$ using the corresponding measured widths. Note that the interpretation of the loss structure also holds for our measurements below $17\,\text{G}$, for which $a_{\text{bg}}$ takes negative values (*c.f.* inset to Fig. 1(a)). Here the zero crossing is located at magnetic fields below the pole. For the ultra-narrow Feshbach resonances, which are too narrow to experimentally resolve a substructure, we identify the center of the single loss feature with the resonance pole. Finally, our results for all extracted positions $B_0$ and widths $\Delta B$ are summarized in Table 1, including the statistical error for the widths and the expected experimental error due to uncertainties in magnetic-field calibration and magnetic-field noise. A comparison to the theoretical values based on latest multichannel calculations [28] shows again a very good overall agreement when taking into account the uncertainty on the theory positions on the order of $0.2\,\text{G}$.

## 5 Conclusion and outlook

In summary, we have explored an alternative technique to detect and characterize Feshbach resonances employing strong correlations of a Mott-insulating many-body state. This allows for the observation of previously unexplored ultra-narrow resonances, which are fully masked by strong three-body loss in more common bulk measurements. The approach is of interest for various elements with complex and overlapping Feshbach spectra e.g. lanthanides like Er and Dy [34–37]. We have provided a detailed analysis of the resonant loss features in the lattice, which show substructure due to the interplay of particle tunneling, interactions, and externally

applied energy offsets. Combined with independent measurements of the resonance widths via Landau-Zener type Feshbach molecule formation, this allows for a full parametrization of the resonances, which is important for improving the accuracy of coupled-channels calculations of molecular potentials [22]. The observed ultra-narrow Feshbach resonances could be suited to realize several proposals in high precision measurements [14–16]. Their location at comparatively low magnetic fields should allow to reach the necessary accuracy, precision, and noise requirements of magnetic fields using passive and active stabilization techniques. Moreover, the substructure observed in the lattice calls for future studies at broader resonances, for which we expect to resolve resonant coupling to higher lattice bands [38, 39].

# Acknowledgements

We are indebted to R. Grimm for generous support. We thank P. S. Julienne for fruitful discussions.

**Funding information** F. M. acknowledges support from the Carl-Zeiss foundation and is indebted to the Baden-Württemberg-Stiftung for the financial support by the Eliteprogramm for Postdocs. We gratefully acknowledge funding by the European Research Council (ERC) under Project No. 278417 and the Austrian Science Foundation (FWF) under Projects No. I2922-N36 and No. Z336-N36.

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
