# Peer review of "Mott-Insulator-Aided Detection of Ultra-Narrow Feshbach Resonances"

_SciPost Physics, doi:SciPost Phys. 5, 055 (2018)_

## Round 1 · Referee Report · Anonymous · 2018-9-27

Strengths

1-great idea that brings a now toolbox to probing atom scattering;
2-new way to study scattering in correlated quantum status in lattices;
3-will lead to new ways for spectroscopy;
4-very clear presentation;

Weaknesses

none

Report

This is a very nice innovative paper, which will give a new impulse for studying interacting quantum matter. Most intertestingly is shows a new way of studying bound states in atom-atom scattering which are usually not accessible, and gives a clean demonstration of how to measure very narrow Feshbach resonances. The presented approach to investigate atom scattering is of interest especially for atoms with complex and overlapping Feshbach spectra like dipolar lanthanides like Er. The presented research will also lead to important input for improving the accuracy of coupled-channels calculations of molecular potentials.

Requested changes

1- Figure 2: use same colors and symbols for the specific resonances in both graphs

---

## Editorial Decision

published